# Formation of bridgmanite-enriched layer at the top lower-mantle during magma ocean solidification

Longjian Xie [1,2]*, Akira Yoneda[1], Daisuke Yamazaki[1], Geeth Manthilake [3], Yuji Higo[4], Yoshinori Tange [4], Nicolas Guignot[5], Andrew King[5], Mario Scheel[5] & Denis Andrault [3]

Thermochemical heterogeneities detected today in the Earth's mantle could arise from ongoing partial melting in different mantle regions. A major open question, however, is the level of chemical stratification inherited from an early magma-ocean (MO) solidification. Here we show that the MO crystallized homogeneously in the deep mantle, but with chemical fractionation at depths around 1000 km and in the upper mantle. Our arguments are based on accurate measurements of the viscosity of melts with forsterite, enstatite and diopside compositions up to ~30 GPa and more than 3000 K at synchrotron X-ray facilities. Fractional solidification would induce the formation of a bridgmanite-enriched layer at ~1000 km depth. This layer may have resisted to mantle mixing by convection and cause the reported viscosity peak and anomalous dynamic impedance. On the other hand, fractional solidification in the upper mantle would have favored the formation of the first crust.

[1] Institute for Planetary Materials, Okayama University, Misasa, Tottori 682-0193, Japan. [2] Bayerisches Geoinstitut, University of Bayreuth, 95440 Bayreuth, Germany. [3] Laboratoire Magmas et Volcans, Université Clermont Auvergne, CNRS, IRD, OPGC, F-63000 Clermont-Ferrand, France. [4] Japan Synchrotron Radiation Research Institute, 1-1-1 Kouto, Sayo, Hyogo 689-5198, Japan. [5] Synchrotron SOLEIL, Gif-sur-Yvette, France. *email: ddtuteng@gmail.com

The possibility that a magma-ocean (MO) induced a primordial chemical stratification has major implications for the mantle state and its dynamics over the Earth's history. For example, it could have induced large-scale provinces atop the core-mantle boundary[1] or a basal MO that would have taken several billion years (Ga) to crystallize[2]. There are geochemical arguments for the persistence of primitive reservoirs, based on the isotopic composition of rare gases in some oceanic island basalts[eg 3] and isotopic differences between the available mantle sources and various chondritic components[4]. The geochemical arguments, however, remain subject to discussions and may be insufficient to refine the complete scenario of MO crystallization.

The mechanism of MO crystallization has been modeled in the past and a key parameter, besides the global cooling rate, appears to be the vertical profile of melt viscosity[5]. Unfortunately, available experimental data are limited to 13 GPa and 2500 K and the first-principles and empirical molecular dynamics simulations present a large discrepancy. For example, viscosities differing by a factor of 50 were reported at the lowermost-mantle P-T conditions of 120 GPa and 4000 K[6,7]. First-principles molecular dynamics (FPMD) calculations should be more robust than empirical molecular dynamics simulations, because of absence of assumption about the charge density. They provide viscosity values within a factor of 2 or 3 of experimental data obtained at low pressures and may have an advantage for simulations at very high pressures[7–9]. However, experimental measurements are critically needed to confirm calculations and refine viscosity values, especially at lower mantle P-T conditions.

In this study, we measure viscosity of melts with forsterite, enstatite, and diopside compositions up to ~30 GPa by in-situ falling sphere viscometry. The viscosity of silicate melts shows complex pressure dependence at least up to 30 GPa. With the measured viscosity, we model the mechanism of the MO solidification. It suggests the formation of a bridgmanite-enriched layer at the top of the lower mantle upon MO cooling.

## Results and discussions

**Experimental conditions**. We performed in-situ falling sphere viscometry in a multi-anvil apparatus coupled with intense X-ray beams generated by the SPring-8 and Source optimisée de lumière d'énergie intermédiaire (SOLEIL) third generation synchrotron sources (see Methods section). We used a relatively large beam (about $2 \times 2$ mm) to record the falling path of a small rhenium sphere in the liquid silicate using an ultra-fast camera (Fig. 1), and a collimated beam ($50 \times 200$ μm) to characterize the sample mineralogy and determine the pressure by X-ray diffraction. The complete fall through the ~1 mm long sample is achieved in less than 1 s, which is sufficiently short to avoid the chemical reaction between the Re-sphere and the molten silicate (Fig. 1c). The falling-sphere terminal velocity yields the melt viscosity based on the Stokes' law (see Methods section).

Major limitations encountered in previous works of the same type were technical difficulties to perform the ultrahigh temperatures (more than ~2500 K) that required to melt the silicate phases entirely and the difficulties to measure the extremely low viscosity of silicate melts at high pressure, requiring very fast radiographic measurements[10–12]. By using a new type of furnace made of boron-doped diamond[13] and ultra-fast camera (frame rate reaches 1000 f/s), we could perform viscosity measurements up to 30 GPa and 3250 K. We investigated the viscosity of melts with compositions similar to major mantle minerals, namely forsterite ($Mg_2SiO_4$, Fo), enstatite ($MgSiO_3$, En), and diopside ($CaMgSi_2O_6$, Di). Measurements have been performed slightly above the melting temperatures (see Methods section, Supplementary Table 1).

**Viscosity measurements and modeling**. Our new results fall in good agreement with previous works[10,12,14–16] performed at lower pressures (Fig. 2). Along the liquidus, viscosities present a complex evolution with pressure for the three difference liquid compositions investigated. Viscosity is a thermally activated process that can be modeled based on the Arrhenius equation. Because our measurements were all performed at temperatures relatively close to the liquidus, we first assume an activation process against a dimensionless temperature, which is obtained by normalizing the experimental temperature to the melting temperature of the specimens at a given pressure:

$$\eta(P, T) = \eta_0 \exp\left(\frac{E_a(P)}{kT}\right) = \eta_0 \exp\left(\frac{E_a^*(P)}{(T/T_m)}\right) = \eta_0 \exp\left(\frac{E_a^*(P)}{T^*}\right) \tag{1}$$

where $\eta_0$ is a scaling factor; $k$ Boltzmann constant; $T$ absolute temperature; $P$ pressure, $T_m$ melting temperature at pressure $P$; $E_a$ activation enthalpy; $T^*$ dimensionless temperature ($T/T_m$); $E_a^*$ dimensionless form of the activation energy. At the liquidus temperature, $T^*$ equals 1 and we obtain:

$$\ln(\eta) = \ln(\eta_0) + E_a^*(P) \tag{2}$$

An accurate pressure dependence of $E_a^*$ can be determined based on the viscosity profile along the liquidus (Fig. 2). Our viscosity data for Fo, En, and Di melt compositions suggest that the pressure dependence of $E_a^*$ can be fitted using third order polynomial fit, at least up to 30 GPa (Supplementary Table 3, Supplementary Fig. 5):

$$\eta(P, T) = \eta_0 \exp\left(\frac{c_0 + c_1 P + c_2 P^2 + c_3 P^3}{T^*}\right) \tag{3}$$

We note that the viscosity profiles can also be fitted using two linear sections (Supplementary Fig. 2, Supplementary Table 3). Based on Eq. (3), we can now recalculate the viscosity of the end-member melts at any temperatures and, in particular, along isotherms.

All of Fo, En, and Di compositions show a weak and complex pressure dependence along isotherms (Fig. 2). Our experimental results are quite consistent (within one order of magnitude) with FPMD predictions, especially for En and Fo composition (Fig. 2). Our results for Di are also consistent with experimental determinations of oxygen and silicon self-diffusions in Di melt[17]. En melt and, to a lesser extent, Di melt show a negative pressure dependence in some pressure ranges. Such an anomalous behavior was also reported in a basalt and another silicate melt[18], based on both FPMD simulation[19] and experimental measurements[20,21]. The negative pressure dependence is due to either the Si–O bond weakening by the pressure-induced bending of the Si–O–Si angle[21,22] or possibly the increasing concentration of five-fold Si–O coordination species[23,24]. The complex pressure dependence correlates nicely with the mechanisms of silicate melt densification described previously[22,25] (details in Supplementary Note 1 and Supplementary Fig. 4).

**Extrapolation of melt viscosity to deep lower mantle conditions**. The knowledge of the dependence of the melt viscosity along mantle isotherms enables the refinement of the true activation enthalpy and its pressure dependence ($E_a$ in Eq. (1); Supplementary Fig. 5). The refined $E_a$ values at room pressure are $100 \pm 20$ and $159 \pm 10$ kJ mol$^{-1}$ for Fo, and En melts, respectively, in agreement with previous FPMD predictions[7,8]. Di melt presents a relative large $E_a$ ($230 \pm 30$ kJ mol$^{-1}$), which is consistent with diffusion experiments ($268$ kJ mol$^{-1}$)[17] but larger than FPMD prediction ($148 \pm 5$ kJ mol$^{-1}$)[9]. Further work is needed to solve

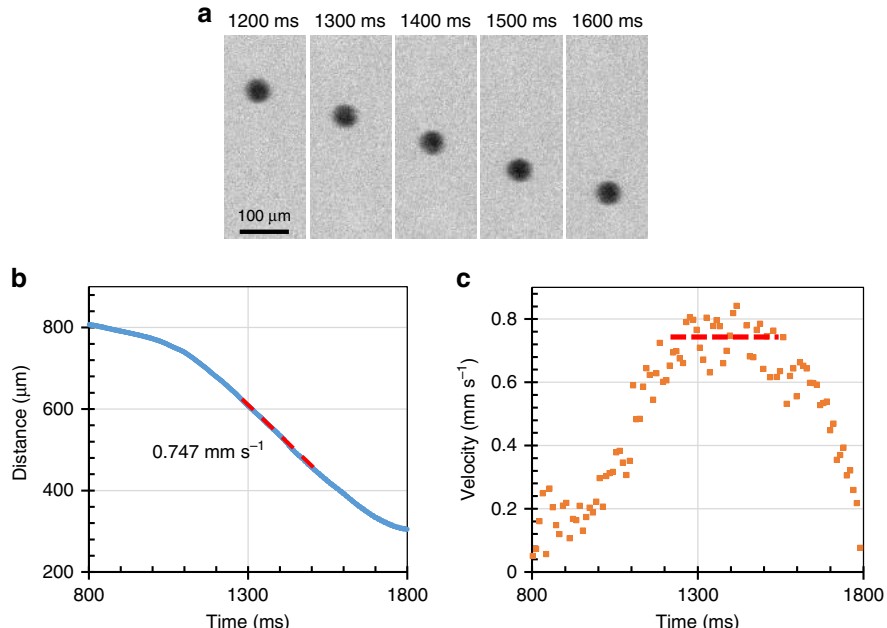

**Fig. 1 Experimental observation of the falling sphere. a** Sequential radiographic images recorded at ~24 GPa and ~2873 K during the fall of a Re-sphere of ~65 μm diameter (Run MA24). **b** Position of the sphere as a function of time in Run MA24. The sphere position was fitted by a Gaussian function in each X-ray radiographic image (blue symbol). The melt viscosity can be calculated from the terminal velocity (red dashed line) using Eq. (4). **c** Velocity/time plot of the sphere in Run MA24, using a sampling time of 10 ms. The red dashed line is a best fit through the data points located on the "velocity plateau" corresponding to the terminal velocity.

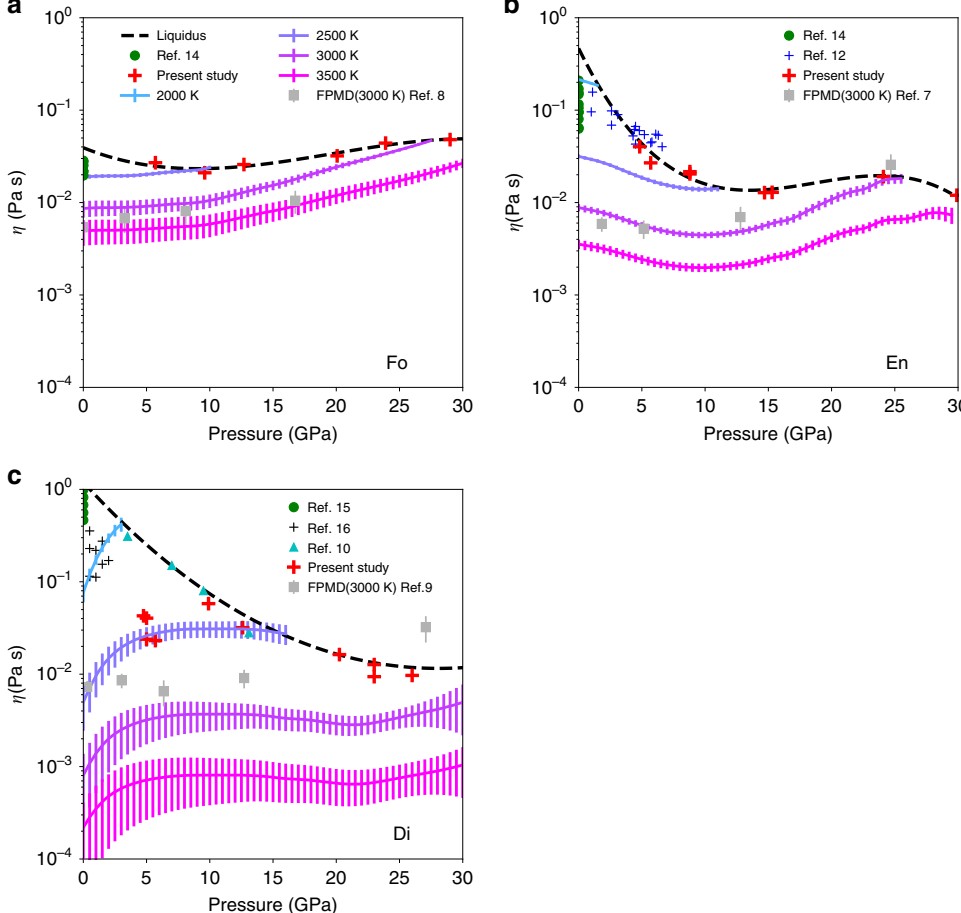

**Fig. 2 Viscosities of silicate melts under pressure. a–c** Fo, En, and Di composition, respectively. We report our experimental data as red crosses, whose temperatures are shown in Supplementary Fig. 2 and Supplementary Table. 1. Dashed black lines are viscosities along liquidus. Colored lines are viscosities recalculated along isotherms with their 1σ standard deviation.

**Table 1 Model for the activation enthalpy (Supplementary Fig. 5).**

| Composition | $\eta_0$ (Pa s) | $a$ (kJ mol$^{-1}$ GPa$^{-1}$) | $b$ (kJ mol$^{-1}$) |
|---|---|---|---|
| Mg$_2$SiO$_4$ | $2.3\ (12) \times 10^{-4}$ | 1.64 (7) | 90 (1) |
| MgSiO$_3$ | $7.63\ (484) \times 10^{-5}$ | 1.14 (19) | 141 (3) |
| CaMgSi$_2$O$_6$ | $3.17\ (685) \times 10^{-8}$ | 0.49 (16) | 253 (3) |

this discrepancy. However, it will not affect the conclusion of this article significantly because the amount of diopside is less than 2% in relevant mantle compositions (Supplementary Table 4) and only extrapolations to the highest mantle pressures could be affected by significant uncertainties, while viscosity at moderate pressures are satisfactorily constrained by the present results.

Linear fits (Table 1) enable the extrapolation of the melts viscosity towards the very deep mantle using the Arrhenius equation ($\eta(P,T) = \eta_0\exp((a\ P + b)/T^*)$). They average the complex pressure-dependence of viscosity at low pressures. Still, the fitted $E_a$ values remain within 10% of the experimental $E_a$ (Supplementary Fig. 5).

**Viscosity of magma ocean.** To estimate the viscosity of silicate melts with compositions relevant to the deep mantle, we now apply the Adam-Gibbs mixing theory. It states that the logarithm viscosity of a complex system can be expressed well as a linear combination of logarithm of the viscosities of end-members[26] (Eq. (10) in Methods section). Thus, we used end-member melts of Fo, En, Di, and anorthite (An, CaAl$_2$Si$_2$O$_8$)[27] to calculate the viscosity, and its dependence with pressure and temperature, of MOs consisting of peridotitic KLB-1 and chondritic-type compositions (Supplementary Table 4). Because water has little effect on a completely depolymerized, high temperature magma viscosity[28,29], we only consider dry MOs. For more accuracy, the pressure-dependence can be modeled using either experimental constraints (Fig. 2) or Ahrenius fits (Supplementary Fig. 5), for lower and higher than 30 GPa, respectively. It appears that the MO viscosity is controlled by its two main chemical components: Fo and En. Our calculations for KLB-1 composition (Fig. 3a) is roughly compatible with available measurements[30]. Interestingly, viscosity profiles present a local minimum at depths around 300–400 km along MO adiabats and a local maximum at ~660 km along the liquidus. Also, the viscosity of KLB-1 is found to be slightly lower than that of the chondritic mantle along their respective liquidus temperatures.

**Major parameters for modeling of magma ocean solidification.** Before a MO behaves like a solid at a crystal fraction higher than ~60%[31], the progressive crystallization could have induced some fractional crystallization. Its occurrence, or not, depends on the competition between the forces favoring the gravitational sedimentation or the suspension of the solid grains in a turbulently convective MO[5]. Above a critical diameter, crystals precipitate at the bottom of the MO. Therefore, the value of crystal/critical diameter ratio (Rcc, see Methods section) is an indicator on whether the MO crystallization occurs with fractional solidification (Rcc > 1) or at chemical equilibrium (Rcc < 1). Fractional solidification is favored by low MO viscosity, low heat flux and large density contrast between crystals and melt.

To model the mechanism of MO solidification, it is necessary to determine the change of Rcc value during MO cooling. For this, preliminary definitions are needed. We define the MO-bottom as the higher mantle depth where the rheological transition ($T_{Rheo}$) already occurred upon MO cooling (see Methods section). The MO-bottom defines the temperature

profile in the entire MO, based on the $T_{Rheo}$ anchor point coupled to the adiabatic temperature gradient in the MO. The calculated potential surface temperature is used to determine the heat flux through the MO, considering loss of heat by thermal radiation (Supplementary Fig. 7a, b). Then, we define the crystallization zone as the range of mantle depths where solid and melt coexist. This region extends from the MO-bottom to the shallower intersection between the adiabatic temperature gradient in the mushy zone and the liquidus profile. Significant uncertainties remain about its thickness, because of the effect of the latent heat of fusion on the adiabatic temperature gradient between the solidus and the liquidus. We consider different assumptions for its thickness below. In all cases, we calculate an averaged melt viscosity and an averaged solid-melt density contrast within the crystallization zone.

To model the solid-melt density contrast, we consider a range of Fe solid-liquid partition coefficient ($K_{Fe}$) from 0.2 to 0.6 (see ref. [32] and references therein) and first liquidus phases that change with MO depth (see Methods section). Higher $K_{Fe}$ favors higher density contrast of bridgmanite over liquid. Upon MO crystallization, the averaged density contrast first increases due to higher bridgmanite density, compared to the melt (Supplementary Fig. 8e, f). Then, it decreases above ~1500 km when, within the upper part of the crystallization zone, the first liquidus phase changes from bridgmanite to majorite at ~660 km and majorite to olivine at ~450 km. It finally increases again at low MO-bottom depth, due to high olivine density at shallow mantle depth. These variations cause a peak and a valley at ~1500 km and ~450 depths, respectively.

As long as a MO remains fully molten at shallow mantle depth, crystals experience a life cycle of nucleation-growth-dissolution due to turbulent vertical convection. With a short residence time of about one week in the crystallization zone[5], grain growth is insignificant and crystal size is controlled by nucleation processes. Later, the crystallization zone reaches the Earth's surface, that is, the whole MO is the crystallization zone, making the crystal lifetime considerably longer. Some grains can survive and grow until the MO is entirely solidified. In this regime, crystal size is controlled by grain growth (Ostwald ripening, see Methods section). The depth when the controlling mechanism switches, depends on the definition of the crystallization zone before the fully molten layer disappeared. To check the robustness of our conclusions, we considered three different situations: the crystallization zone extends (1) in the entire MO, (2) up to 1000 km above the MO bottom, and (3) up to the intersection between the MO liquidus and the adiabatic temperature profile, neglecting the role of the latent heat of fusion. Within these three assumptions, Ostwald ripening increases the grain size significantly from the onset of MO solidification, when the MO-bottom reaches 1000 km depth (in agreement with previous reports considering the latent heat of fusion within the crystallizing zone[33]) and when the MO bottom reaches ~700 (peridotitic MO) or ~150 km (chondritic-type MO) depths, for situations 1, 2, and 3, respectively. To enforce the robustness of our conclusion, we also calculated Rcc value without any grain growth. In reality, the mechanism of MO cooling should be close to situation (2).

**Mechanisms of magma ocean solidification.** We now model the progressive cooling of a MO with an initial depth of 2900 km, as possibly occurred on Earth after the major Moon Forming Impact[34]. We based our discussion on the Rcc value, which indicates a high probability of fractional solidification when it becomes higher than unity. For a chondritic-type MO, Rcc value is found larger than unity in a range of depths around 1000 km

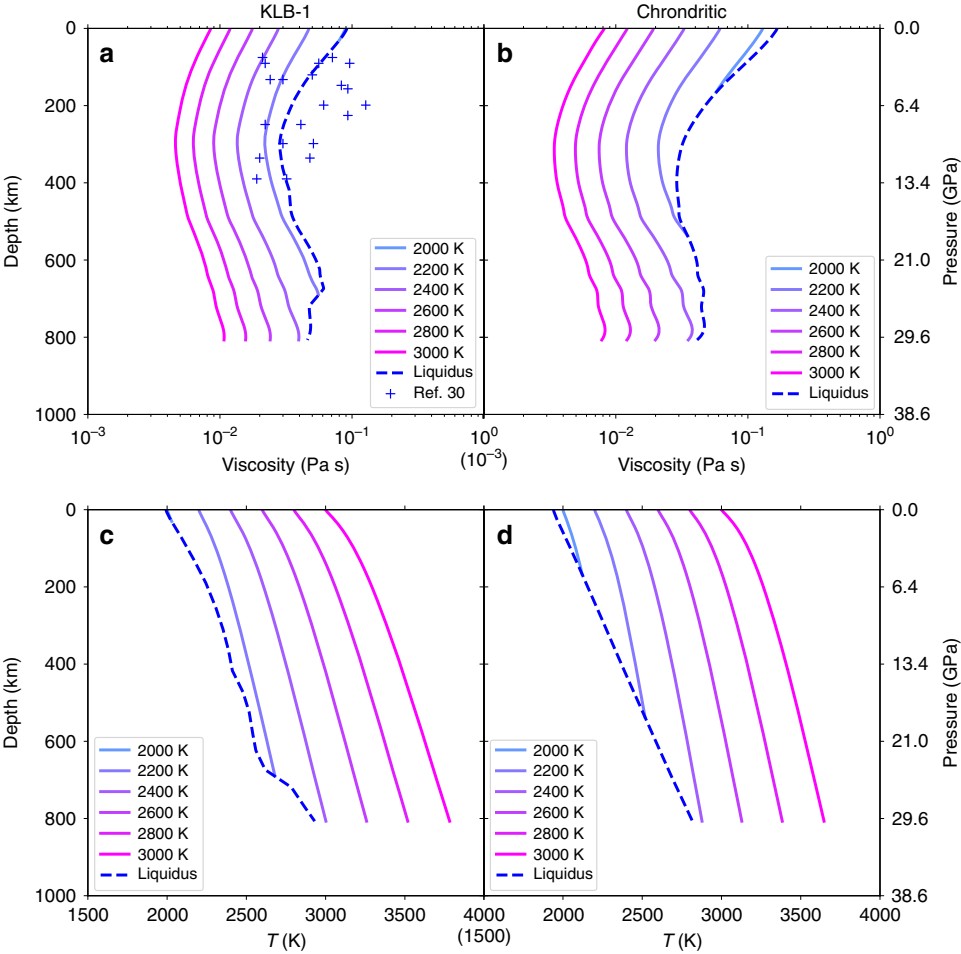

**Fig. 3 Change of magma ocean viscosity with depth. a**, **b** We report MO viscosities with KLB-1 (left) and chondrite-type (right) compositions for potential surface temperatures from 2000 to 3000 K. Solid and dashed lines are calculated along mantle adiabats and liquidus profiles, respectively. **c**, **d** Solid and dashed lines represent adiabatic[63] and liquidus[55,56] profiles, respectively, for molten KLB-1 and chondrite-type mantle compositions.

depth (Fig. 4) even when using the less favorable parameters of $K_{Fe}$ equal 0.2 and a maximum MO heat flux, thus ignoring the possible impact of a blanketing atmosphere (Supplementary Fig. 11b). For KLB-1 composition, fractional solidification would also occur around similar depth regardless of the value of $K_{Fe}$, if the blanketing effect is larger than 20% ($C_f$ smaller than 0.8) (Fig. 4a and Supplementary Fig. 11a–c). At such mantle depths, Rcc values are weakly dependent on the effect of grain growth by Ostwald ripening, except if the crystallization zone extends from the surface to deeper than 1000 km. In this case, the fractional crystallization would be even more likely (Fig. 4 and Supplementary Fig. 10). Altogether, we did a very conservative calculation of Rcc in the present study (in particular for heat flux estimations, see Methods section). Therefore, fractional solidification with sedimentation of bridgmanite grains should occur around 1000 km depth for any MO composition between chondritic-type and peridotite.

We also investigate the possible sedimentation of ferropericlase (Fp), which is the liquidus phase below 35 GPa, at least for the peridotitic composition[35] (Supplementary Fig. 11c, d). Nevertheless, due to its lower density compared to bridgmanite, Fp yields much smaller Rcc values, especially in absence of significant Ostwald ripening. Thus, Fp tends to remain suspended in the melt.

**Implications for the state of the Earth's mantle**. The sedimentation of bridgmanite grains implies the formation of a

bridgmanite-enriched layer at depths around ~1000 km. It also implies an enrichment of a shallow MO toward the peridotitic composition, with a higher MgO content compared to the primitive chondritic-type mantle. Such chemical fractionation remained only partial, however, as evidenced by the available geochemical constraints[36,37]. Such an early bridgmanite-enriched layer may have survived until present, despite mantle convection, as suggested recently based on geodynamical simulations[38]. Within this assumption, the bridgmanite-enriched layer could cause the viscosity increase reported at mantle depths between 660 and 1500 km[39], which appears to impede the dynamic flow in this mantle region[40].

In a shallow MO, the Rcc values present a major increase above unity when the controlling mechanism for grain size switches from nucleation to grain growth. It corresponds to a major increase of grain size due to Ostwald ripening, favoring fractional crystallization. This effect is more pronounced when the crystallization zone is thick when approaching the Earth surface. The solidification of the upper mantle with fractional solidification of garnets and olivine could have triggered the formation of a proto-crust at the surface of the Earth.

## Methods

**Experiments at high pressures and high temperatures**. Melt viscosities were measured by in-situ falling sphere method in a Kawai-type multi-anvil apparatuses installed at synchrotron-based BL04B1 (SPring-8) and Psiché (SOLEIL) beamlines. We used cubic WC anvils with 26 mm edge and 4 mm truncation edge

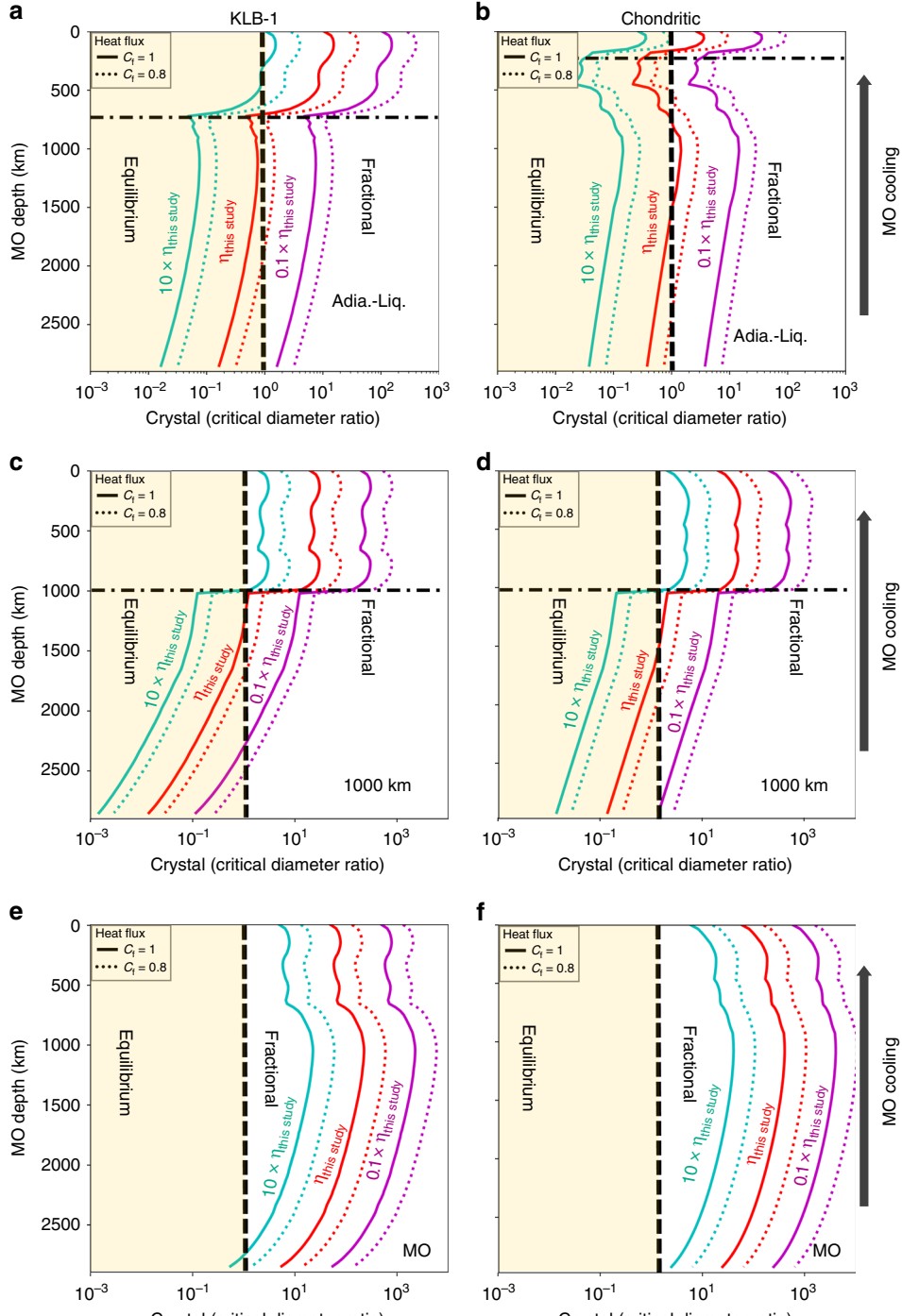

**Fig. 4 Crystal/critical diameter ratio as a function of magma ocean depth.** *Rcc* parameter calculated for KLB-1 MO (**a**, **c**, **e**) and chondritic-type MO (**b**, **d**, **f**) using bridgmanite as solid phase and a solid-melt Fe partition coefficient of 0.6. We considered MO viscosities of 10 (green), 1 (red), and 0.1 (blue) times the MO viscosity determined in this study. Solid or dotted profiles correspond to no blanketing atmosphere (*Cf* = 1) or an atmosphere reducing the effective surface temperature by 20% (*Cf* = 0.8, see Eq. (16)), respectively. Zones colored in yellow (Rcc lower than 1) or white (Rcc higher than 1) indicate MO expected to solidify at chemical equilibrium or through fractional solidification with sedimentation of bridgmanite, respectively. The horizontal black dashed-dotted line marks the MO-bottom depth when crystallization starts at the surface of the MO. Upper, middle and lower frames consider a crystallization zone that extends from the MO bottom to (**a**, **b**) the intersection of adiabatic and liquidus profiles (Adia.-Liq.; see situation (3) in the main text), (**c**, **d**) 1000 km above MO bottom, (**e**, **f**) MO surface, respectively.

length to generate pressures up to ~30 GPa, corresponding to ~800 km depths in the Earth's mantle. Pressure medium was $Cr_2O_3$-doped MgO octahedron of 10-mm edge length with edges and vertexes truncated (Supplementary Fig. 1a). The polycrystalline sample was loaded in a graphite capsule. Thermocouple ($W_{97}Re_3$-$W_{75}Re_{25}$) was placed below the graphite capsule. We used MgO mixed with 10 wt% diamond (with ~1 μm grain size) as internal pressure marker, based

on the *P*-*V*-*T* equation of state (EoS) of MgO[41]. The role of the diamond powder is to prevent the MgO grain growth. To determine accurate sample pressure, we used an MgO volume recorded as close as possible from the thermocouple. Pressure uncertainty is estimated to be less than 1 GPa, including the propagation of uncertainties on determination of MgO volume, temperature (see below) and the EoS itself.

To ensure laminar flow during the fall of the Re sphere in the low-viscosity silicate melt, a Re sphere of ~70 µm diameter was placed near the top and at the center of the sample. The spheres were prepared from stripes of 25 µm thick Re foil by applying a flash current at 100 V. The Re stripes were immerged in liquid nitrogen to prevent oxidization and enhance the quenching rate. The sphere diameters were measured using a field emission scanning electron microscope with an accuracy better than ±2 µm. The recovered samples were confirmed to be free from any chemical reaction between the Re spheres and the silicate melt (Supplementary Fig. 1c).

As heater, we used graphite or boron doped diamond (BDD) at pressures lower or higher than 8 GPa, respectively. The BDD heater can generate temperature as high as ~4000 K with highly X-ray transparency, which is ideal to perform in-situ falling sphere viscometry at high pressures in multi-anvil apparatus[13]. The accurate determination of the falling-sphere terminal velocity in low-viscosity melts requires the use of ultra-fast camera (1000 fps) coupled with synchrotron X-ray radiography[11,12].

Experimental runs were typically performed as follow: Compression to the target pressure at 300 K, progressive heating to 1273 or 1773 K (depending on the sample pressure) to determine the power-temperature relation and record the relative positions of the sample and the pressure marker, before we conducted a fast heating (2–4 s) up to the maximum target temperature. The target temperature was set a couple 100 K above the liquidus temperature (from ref. [42] and references therein for Fo and En and from ref. [43] and ref. [44] for Di below and above 17 GPa, respectively). Because the thermocouple usually broke during the fast heating ramp, the ramp was monitored based on the power-temperature relation determined previously (see below our simulations of the sample behavior during the fast heating ramp). This procedure yields a temperature uncertainty of ~30 K, or ~150 K, with, or without, a thermocouple reading, respectively. Then, after we observed the sphere falling, we kept the power constant and measured the sample pressure again.

**Temperature gradients**. To estimate temperature gradients, we recorded diffraction patterns at different positions in the MgO pressure marker (positions noted P1, P2, and P3 in Supplementary Fig. 1d). Under the assumption of a negligible pressure gradient inside heater, the difference in MgO volumes can be attributed to a temperature difference. At a thermocouple temperature of 1273 K, the resulted temperature difference is less than 60 K between the thermocouple and center of capsule, and less than 10 K between the center and the top of capsule.

**Simulations of the sample behavior during the final heating procedure**. To prevent a chaotic fall of the Re-sphere in a partially molten sample, the final heating step consisted in a ramp of fast heating. For a liquidus temperature expected 2500 K, for example, we typical set a heating ramp from 1773 to 3000 K within a duration of 2 s (i.e. ~600 K/s). To model the sample behavior during this ramp, we conducted finite element simulation using the COMSOL™ software. To simplify the sample geometry without losing the essence of the critical part inside the heater, the octahedral shape of the pressure medium was modeled as a cylinder. The sample thermal conductivity was assumed to be 50 and 2.5 W/(mK) before and after melting, respectively.

Our calculations show that the effect of the latent heat of fusion turned out to be negligible. The temperature gradients in the capsule are within ~20 K, in good agreement with our estimates based on the MgO equation of state. The measurement of the falling sphere velocity is performed at an "overshoot temperature" above the liquidus, which can be estimated from the time consumed to reach the terminal velocity. For the sample MA24, the terminal velocity is achieved in less than ~0.3 s after the onset of the sphere fall. With a heating ramp of 600 K/s, this implies that the terminal velocity is achieved at about 180 K above the liquidus temperature, a temperature gap similar to the uncertainty in the temperature determination using the relation between the furnace-power and the sample temperature.

**Viscosity calculation**. The falling sphere velocity was determined based on the recording of high quality images from the fast camera (1000 fps) installed at the synchrotron facilities. The falling speed first increases before it reaches a constant (terminal) velocity (Fig. 1). The distance interval where terminal velocity was reached, was determined through the velocity-time diagram (Fig. 1c). The terminal velocity ($v_s$) corresponds to the state where viscous forces are equilibrated with the gravitational force. The Reynold numbers of all our experiments (0.01–0.1) are far smaller than 1, which is in the laminar flow regime. Therefore, the terminal velocity yields the melt viscosity ($\eta$), based on the Stokes law:

$$\eta = \frac{2gr_s^2(\rho_s - \rho_m)W}{9v_sE} \quad (4)$$

$$W = 1 - 2.104\left(\frac{r_s}{r_c}\right) + 2.09\left(\frac{r_s}{r_c}\right)^3 - 0.95\left(\frac{r_s}{r_c}\right)^5 \quad (5)$$

$$E = 1 + 3.3\left(\frac{r_s}{h_c}\right) \quad (6)$$

where $r_s$, $\rho_s$, $\rho_m$, and $g$ correspond to sphere radius, sphere density, melt density, and acceleration due to gravity, respectively. $W$ and $E$ are correction factors accounting for the presence of walls and end in a sample container of radius $r_c$ and height $h_c$[45]. The radius and the density of the Re spheres were corrected for the effect of pressure using the EoS of Re[46]. The density of Fo, En, and Di melts were calculated based on the available EoS[47–49].

**Error analysis and reproducibility**. We conducted Monte Carlo simulations to evaluate the propagation of experimental uncertainties on pressure, temperature, terminal velocity and sphere size. Gaussian distribution of experimental uncertainties was assumed. The sampling number was 10,000. The results for Run S3219 (En, 24.1 GPa, 2836 K) are presented as an example (Supplementary Fig. 3). Even though the relative uncertainties of pressure and temperature are larger (~3.6%) than those of terminal velocity (0.5%), their contribution to the final viscosity is 1 order of magnitude smaller. This is because the density contrast between sphere and melts is not sensitive to pressure and temperature. The main source of uncertainty for the final determination of the melt viscosity is caused by the uncertainty of 2.9% on the sphere diameter, which is elevated to a quadratic power in the expression of viscosity. The total error on viscosity is within 6%, with an almost Gaussian distribution (Supplementary Table 1).

Reproducibility of our measurements was checked by performing repeated experiments at similar pressures, temperatures and with different sphere sizes (such as run S3170 and S3171, S3172 and S3175, S3257 and S3260 in Supplementary Table 1). The difference between repeated experiments remains within 6%, which is consistent with the estimated viscosity error.

**Viscosity of peridotite composition in MO**. Bottinga and Weill (1972)[50] proposed that the logarithm viscosity of a multi-components melt at super-liquidus conditions can be satisfactorily expressed as a linear function of the logarithm viscosity of the end-member compositions over a restricted composition interval (for example $SiO_2$ mole content from 30–50%). This model is supported by the Adam-Gibbs theory, because viscosity can be expressed as a function of the configuration entropy ($S^{conf}$)[26]:

$$\eta = A_e \exp\left(\frac{B_e}{TS^{conf}}\right) \quad (7)$$

$$S^{conf}(T) = \sum x_i S_i^{conf}(T) + S_{mix} \quad (8)$$

$$S_{mix} = -nR\sum x_i \ln x_i \quad (9)$$

where $n$ is the number of entities exchanged per formula unit. When temperature is near liquidus or higher, $S_{mix}$ is very small and negligible. Therefore, we can use linear combination of logarithmic viscosity. This model was experimentally confirmed for the Ca–Mg exchange in molten garnets and pyroxenes[51] and for Na–K exchange in alkali-silicates[52]. In our case, we model the viscosity of mantle melts with KLB-1 or chondritic-type compositions (Supplementary Table 4) based on four end-members, Fo, En, Di, and An:

$$\ln(\eta_{mantle}) = f_{Fo} \times \ln(\eta_{Fo}) + f_{En} \times \ln(\eta_{En}) + f_{Di} \times \ln(\eta_{Di}) + f_{An} \times \ln(\eta_{An}) \quad (10)$$

where $f_i$ are the molar contents of each endmember. Viscosity of Fo, En, Di melts are provided from the present work and viscosity of An was reported from first principle calculation[27]. We chose viscosity function of Fo for both Fo and Fa components. Because Fa component represents less than 8% of the KLB-1 and Chondritic composition and, in addition, viscosities of Fa and Fo converge to the same value (the difference is within ~10%) with increasing pressure (Supplementary Fig. 6). The total error caused by ignoring the Fa component is less than 0.8%.

**Temperature at the bottom of a MO**. In this work, we defined the bottom of the MO as the mantle depth where the rheological transition first occurs, at a crystal fraction of 60%. The temperature at this depth ($T_{Rheo}$) is at an intermediate temperature between the solidus ($T_{sol}$) and the liquidus ($T_{Liq}$). By lack of knowledge, we assume a linear evolution of the degree of partial melting between $T_{sol}$ and $T_{Liq}$ (as in ref. [53]). Therefore, $T_{Rheo}$ equals $0.4 \times T_{Liq} + 0.6 \times T_{sol}$.

For the peridotitic mantle composition, we used $T_{sol}$ and $T_{Liq}$ profiles from ref. [54] and ref. [55], respectively, while for a chondritic-type mantle, we considered the $T_{sol}$ and $T_{Liq}$ from ref. [56] at pressures >8 GPa and from ref. [57] at pressures <= 8 GPa.

**The critical grain diameter for sedimentation in a convecting MO**. The critical diameter is the maximum size of crystal that the MO convection can suspend. In this case, the viscous dissipation equals the total heat loss rate from the MO[5]

$$W = \frac{\alpha g L F A}{c_p} \quad (11)$$

where $L$, $A$ are the MO depth and surface, respectively; $F$ the heat flux through the MO; $\alpha$ the averaged thermal expansion coefficient of MO; $W$ the viscous

dissipation energy.

$$W = v_s g \int_{L_{CZT}}^{L} \Phi \Delta \rho \, dV \tag{12}$$

$v_s$ is the relative velocity between crystal and melt, $g$ gravity acceleration, $\Delta\rho$ the averaged density contrast between melt and crystal, $V$ the volume of crystallization zone, $\Phi$ the crystal fraction, $L_{CZT}$ is the depth of top surface of crystallization zone.

Assuming linearly increase of $\Phi$ with depth from 0 at top of crystallization zone to ~0.6 at the viscous transition. At a given depth (D), $\Phi$ can be expressed as:

$$\Phi = \frac{0.6(D - L_{CZT})}{L - L_{CZT}} \tag{13}$$

$dV$ can be expressed as:

$$dV = 4\pi(R - D)^2 \, dD \tag{14}$$

where $R$ is the Earth's diameter.

Combining Eqs. (11–14), we obtain:

$$v_s g \int_{L_{CZT}}^{L} \frac{0.6(D - L_{CZT})\Delta\rho 4\pi(R - D)^2}{L - L_{CZT}} \, dD = \frac{\alpha g L F A}{c_p} \tag{15}$$

In the early stages of the MO crystallization, the surface temperature of MO ($T_{sur}$) is expected to be more than 2000 K, producing an atmosphere made of silicate rock vapor[58]. In such conditions, the heat flux at the MO surface is estimated by:

$$F = \sigma_{SB}(C_f T_{sur})^4 \tag{16}$$

where $\sigma_{SB}$ is the Stefan–Boltzmann constant and $C_f$ the ratio between the effective surface temperature (the temperature that would produce a given heat flux $F$) and $T_{sur}$. For a moderately opaque atmosphere with $C_f = 0.75$, the heat flux is ~3 times lower than that for $C_f = 1$. $T_{sur}$ is related to the potential temperature ($T_{poten}$) of MO through the scaling law. When $T_{poten}$ is 2000 K, $T_{sur}$ is ~1800 K and 1400 K for hard and soft turbulent convection, respectively[5]. Here, we assume the $T_{sur}$ equals $T_{poten}$. Therefore, $T_{sur}$ is overestimated by 1.1 to 1.4 times.

Combining Eqs. (11), (12) and (16), we obtain:

$$v_s = \frac{\alpha L A \sigma_{SB} \left(C_f T_{sur}\right)^4}{c_p \Phi \Delta \rho V} \tag{17}$$

The overestimation of $T_{sur}$ causes an overestimation of $v_s$ by ~1.4 to 4 times.

When crystals are suspended in the melt, the viscous drag (right part of Eq. 14) balances the buoyancy force (left part of Eq. 14). Assuming a crystal with a sphere shape, we obtain:

$$\frac{4\pi \Delta\rho g \left(\frac{d}{2}\right)^3}{3} = \frac{C_d \rho_l \left(\frac{v_s}{f_\Phi}\right)^2 \pi \left(\frac{d}{2}\right)^2}{2} \tag{18}$$

and thus,

$$dc = \frac{3 C_d \rho_l \left(\frac{v_s}{f_\Phi}\right)^2}{4 \Delta\rho g} \tag{19}$$

where $dc$ is the critical crystal diameter, $\rho_l$ the average density of melt in the crystallization zone, $C_d$ the drag coefficient and $f_\Phi$ the hindered settling function. Because the crystal fraction in the crystallization varies from 0 to 0.6 as a function of mantle depth and time, we consider an average crystal fraction of 30%, which corresponds to a $f_\Phi$ value of 0.15[59]. The drag coefficient depends on the shape of particle and the Reynolds number:

$$\text{Re} = \frac{\rho_l \left(\frac{v_s}{f_\Phi}\right) dc}{\eta_l} \tag{20}$$

Where $\eta_l$ is the averaged viscosity in the crystallization zone.

For a spherical shape, $C_d$ can be expressed as[60]:

$$C_d = \frac{24}{\text{Re}} + \frac{2.6\left(\frac{\text{Re}}{5.0}\right)}{1 + \left(\frac{\text{Re}}{5.0}\right)^{1.52}} + \frac{0.411\left(\frac{\text{Re}}{2.6 \times 10^5}\right)^{-7.94}}{1 + \left(\frac{\text{Re}}{2.6 \times 10^5}\right)^{-8.00}} + \frac{0.25\left(\frac{\text{Re}}{10^6}\right)}{1 + \left(\frac{\text{Re}}{10^6}\right)} \tag{21}$$

Combining Eqs. (19–21), we can obtain $dc$ numerically. The Reynolds number in MO is less than 20 when grain size in MO equals critical size (Supplementary Fig. 7e, f). Thus, $C_d$ is larger than 1. Because we overestimate the heat flux by ~1.4 to 4 times, the calculated value of $dc$ is at least overestimated by ~2 or 14 times.

**The diameter of crystals in the MO.** The controlling mechanism for crystal size in the MO is nucleation or grain growth before or after, respectively, a fully-molten layer disappears at the shallow mantle depths. Before a fully-molten layer dis-appears at the shallow mantle depths, crystals nucleate, grow and dissolve on the course of their vertical movement in the convecting MO. The nucleation size can be estimated using the following equation[5]

$$d_{nucl} \approx 0.001 \left(\frac{\sigma_{app}}{0.02 \, \text{J m}^{-2}}\right) \left(\frac{D}{10^{-9} \text{m}^2 \text{s}^{-1}}\right)^{1/2} \left(\frac{\mu_0}{10 \, \text{m s}^{-1}}\right)^{-1/2} \tag{22}$$

where $\sigma_{app}$ is the apparent surface energy, $D$ the diffusion coefficient in the melt and $\mu_0$ the convection velocity. $\mu_0$ is correlated to the heat flux[5].

$$\mu_0 = 14 \left(\frac{\alpha g F}{\rho c_p \Omega}\right)^{1/2} \tag{23}$$

where $\rho$ is the averaged melt density of MO and $\Omega$ the angular velocity. On the other hand, the coefficient $D$ in Eq. (22) can be related to the melt viscosity using the Eyring equation[61]:

$$D = \frac{k T_z}{\lambda \eta_l} \tag{24}$$

where $k$ is the Boltzmann constant, $T_z$ the average temperature in the crystallization zone and $\lambda$ the ionic translation distance, for which we used the diameter of oxygen anion (2.8 Å). Finally, using parameters typical of the MO (Supplementary Table 5), we obtain:

$$d_{nucl} \approx 0.001183 \left(\frac{D}{10^{-9} \text{m}^2 \text{s}^{-1}}\right)^{1/2} \left(\frac{\alpha g F}{\rho c_p \Omega}\right)^{-1/4} \tag{25}$$

During the residence time of crystals in the partially molten layer of ~$10^6$ s (roughly one week)[5], crystal growth ($d_{Os}$) due to Ostwald ripening can be estimated as[5]

$$d_{Os} \approx 0.001 \left(\left(\frac{D}{10^{-9} \text{m}^2 \text{s}^{-1}}\right)^{1/3} \left(\frac{\mu_0}{10 \, \text{m s}^{-1}}\right)^{-1/3}\right) \tag{26}$$

This is similar to nucleation size and, thus, the crystal size is not increased substantially by Ostwald ripening. Therefore, the crystal size is mainly controlled by its nucleation size, when the crystallization zone is covered by a fully-molten layer. In such conditions, we can ignore grain growth and Eq. (25) provides a lower limit for the crystal size in the crystallization zone.

$$d_{crystal} = d_{nucl} \tag{27}$$

When the MO temperature is below the liquidus profile at all depths, some crystals will survive and grow until final settling at the MO bottom. Let's assume d$t$ is the time needed to freeze a small depth d$L$ of the MO after temperature drops by $dT$ in the MO and $dT_{sur}$ at the Earth's surface. If we ignore the energy released due to (i) crystallization and (ii) mantle cooling below the MO-bottom ($L$), we obtain:

$$\frac{dt}{dL} = \frac{M_{MO} c_p dT}{FA} \tag{28}$$

$M_{MO}$ is the total mass of a MO extending up to a depth $L$. According to the adiabatic profile in the MO (Fig. 3c, d), $dT > dT_{sur}$. We obtain:

$$\frac{dt}{dL} > \frac{M_{MO} c_p dT_{sur}}{FA} \tag{29}$$

Let's assume $L_{md}$ the MO depth when, upon cooling, its temperature becomes lower than the mantle liquidus at the surface. The residence time of crystals in a MO with depth $L$ ($L < L_{md}$) equals the time for the bottom of the MO to solidified from $L_{md}$ to $L$:

$$t_{residence} > \int_{L}^{L_{md}} \frac{M_{MO} c_p dT_{sur}}{FA} \, dL \tag{30}$$

The average freezing time, calculated using Eq. (30) is ~10 years per kilometer. Under such cooling rate, crystals can grow substantially after their formation by Ostwald ripening. Under a diffusion-controlled mechanism, crystal size is proportional to $t_{residual}^{1/3}$[35,62].

$$d_{crystal} \approx \left(\frac{t_{residence}}{10^6}\right)^{1/3} d_{nucl} \tag{31}$$

The lower limit of crystal size can be expressed as:

$$d_{crystal} = \left(\frac{\int_{L}^{L_{md}} \frac{M_{MO} c_p dT_{surface}}{FA} dL}{10^6}\right)^{1/3} d_{nucl} \tag{32}$$

**Crystal/critical diameter ratio in a crystallizing MO.** We can now define the ratio of crystal/critical grain diameter (Rcc):

$$\text{Rcc} = \frac{d_{crystal}}{d_c} \tag{33}$$

Rcc values higher, or lower, than unity correspond to the grain sedimentation at the bottom of the MO (fractional solidification), or grain suspension in the convective MO (equilibrium solidification), respectively. Since our model overestimates critical size ($d_c$) by ~2 or 14 times and uses a lower limit of crystal size ($d_{crystal}$), it also underestimates the Rcc value and favors equilibrium solidification.

**Density of melt and crystal in MO.** On the course of MO cooling, we considered the possible crystallization of bridgmanite (($Mg_{1-x}Fe_x$)$SiO_3$) or ferropericlase ($Mg_{1-x}Fe_x$)O at pressures above 23 GPa, garnet (($En_{80}Py_{20}$)$_{1-x}Al_x$) between

15 and 23 GPa, and olivine (($Mg_{1-x}Fe_x$)$_2SiO_4$) below 15 GPa. As our results show that crystal fractionation remains limited to some mantle regions, we assume a constant melt composition on the course of the MO solidification. Density of the MO melt was calculated from endmember melt compositions using the ideal mixing model[63]. Iron content in crystals were calculated based on the melt composition and crystal-melt partition coefficients. We considered partition coefficients varying from 0.2 to 0.6, due to remaining experimental uncertainties (see ref. [3] and references therein). To calculate density of bridgmanite, ferropericlase, majorite and olivine, we used an ideal lattice mixing model[64] between end-member compositions with the following EoS: bridgmanite (($Mg_{1-x}Fe_x$)$SiO_3$);[64] ($Mg_{1-x}Fe_x$)O as a solid solution of MgO[37] and FeO;[65] $En_{80}Py_{20}$)$_{1-x}Al_x$ as a solid solution of $En_{80}Py_{20}$[66] and Almandine;[67] Ol ($Mg_{1-x}Fe_x$)$_2SiO_4$) as a solid solution of ($Mg_{0.9}Fe_{0.1}$)$_2SiO_4$[68] and $Mg_2SiO_4$[69].

## Data availability

The authors declare that the majority of the data supporting the findings of this study are available in the paper or supplementary materials. The unpublished data are available from the corresponding author upon request. An example sphere falling videos is available in the supplementary video.

## Code availability

The Monte Carlo simulation and finite element simulation were conducted with a commercial software MATLAB™ and COMSOL™, respectively. The images were analysis using public software Fiji, which is an open source image processing package based on ImageJ.

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

## Acknowledgements
We thank T. Yoshino, F. Xu, E. Boulard, N. Tsujino, H. Gomi, C. Zhao, Y. Zhang, M. Sakurai, V. Jaseem, and C. Oka for their assistance in high-pressure, high-temperature experiments. We thank R. Njul, D. Wiesner, D. Krauße for the help on polishing sample, measuring SEM and Microprobe, respectively. Discussions with E. Ito, M. Kanzaki, A. Suzuki, and C. Wang helped design the project, and with F. Noritake, S. Ohmura, T. Tsuchiya, X. Xue, S. Yamashita, Y. Wang, and D. Dobson improved knowledge of silicate melt. We thank J. Monteux for the discussion on the adiabats of a magma ocean, S. Karato for the discussion on fitting of the experimental data and D.J. Stevenson for the discussion on viscous drag for different flow patterns. We thank M. Izawa for the proof reading of the paper and T. Katsura for suggestions on improving figures. The BDD powder was grinded at the Geodynamic Research Center, Ehime University under the PRIUS program with T. Irifune and T. Shinmei (Project Nos. A48, 2016-A02, 2017-A01, 2017-A21, and 2018-B30). This work was supported by JSPS Research Fellowship for Young Scientists (DC2-JP17J10966 to L. Xie) and Grants-in-Aid for Scientific Research (Nos. 22224008 and 15H02128 to A.Y.). This is a contribution n°383 to the ClerVolc program. The in-situ falling sphere experiments were performed under SPring-8 Budding Researcher Support Program (Nos. 2015A1771, 2016A1651, 2016B1686, 2017B1686, and 2018A1637) and SOLEIL research proposals (20160333, 20170194).

## Author contributions
L.X. and A.Y. designed the project. L.X. planned and performed experiments with D.A., A.Y., G.M., D.Y., Y.H., Y.T., N.G., A.K. and M. S. L.X. did the image analysis with A.K. L.X. performed the data analysis and Monte Carlo simulation. L.X., D.A. and D.Y. developed the homologous scaling model. A.Y. performed the finite element analysis for overshoot of temperature during experiments. L.X., D.A. and A.Y. developed the model of magma ocean solidification. The paper was written by L.X., A.Y. and D.A.

## Competing interests
The authors declare no competing interests.
