## [Peer Review File · Nature Communications]

Reviewers' comments:

Reviewer #1 (Remarks to the Author):

This study reports novel, highly accurate in situ measurements of the viscosity of silicate melts at extremely high pressures and applies the results to the early molten Earth, concluding that crystal/melt segregation in the molten mantle and the associated chemical differentiation was limited to the uppermost part of the lower mantle and the upper mantle. The type of experiments that the authors report are among the most difficult ones to do. This is clearly the strongest part of the study.

The application part could be somewhat better. For example, during crystallization of the shallow parts of the mantle the crystal size is controlled not by nucleation but by Ostwald ripening - the switch in the controlling mechanism occurs when the temperature drops below liquidus at all depths (ref 5). The crystals can grow an order of magnitude larger. This does not change the main conclusions but it does change the confidence level and the wording of the conclusions - in the upper mantle, fractional crystallization not just "could have also occurred" (last sentence of the abstract), but almost certainly occurs, even regardless of the heat flow uncertainties.

For the sake of completeness of error analysis, the authors should use some errors in the formulas describing the critical crystal size, convective velocities, etc, in their theoretical formulas and fig. 4.

The word "entrainment" on line 151 assumes a mechanism of suspension which is not required by formula 11. "Suspension" is physically a more accurate term.

There are a few typos here and there - the paper needs some proofreading.

Reviewer #2 (Remarks to the Author):

The paper by Jie et al. reports new experimental data on pressure variation of viscosity of three key liquids: forsterite, enstatite and diopside. It also presents implications for magma ocean dynamics and cooling.

In overall, the work is interesting, significantly advancing our knowledge of viscous behavior of silicate melts under relevant P-T conditions.

Here are several suggestions/comments for the authors to consider:

a) It may not be true that the calculated melt viscosities differ by factor of 50 or so between different computational studies. Actually, first-principles molecular dynamics (FPMD) simulations are robust and reasonably accurate. FPMD results are within a factor of 2 or 3 of existing experimental data at zero or low pressures for all three (forsterite, enstatite and diopside) liquids studied in this study (please see Ref. 7 and 15).

So, one should not put FPMD study in the same category as empirical model calculations such as Ref 6. The authors should rephrase lines 45 to 48 to avoid this confusion and refer to relevant references. One such reference is: Verma and Karki, First-principles study of self-diffusion and viscous flow in diopside ($\text{CaMgSi}_2\text{O}_6$) liquid, *Am. Mineral.* 97, 2049, 2012.

b) The paper talks about molecular dynamics simulation study in the context of densification mechanism. The reference is missing.

c) It would be helpful to show the first-principles results along 3000 K for comparison in Figure 2 (and Extended Fig. 2). These results are available for each of three melts studied in this paper (see Ref. 7 and 15 for enstatite and forsterite, and Verma and Karki, 2012 for diopside). The measured relatively weak pressure dependence of melt viscosity appears to be consistent with computational trends.

d) Viscosity results are available for multi-component melts (of similar or different) compositions over pressure regimes of upper mantle and shallow lower mantle. Their viscosity might also show modest variations with depth like this study. The authors show (or at least, discuss) some relevant results (Figure 4). Please see the following references:

- Wang et al., Atomistic insight into viscosity and density of silicate melts under pressure. *Nat. Commun.* 5, 3241, 2014.
- Karki et al., Simulation of silicate melts under pressure, *Magma Under Pressure*, (Chapter 16): 419, 453, 2018.

e) Activation enthalpy (Extended Table 3 and Fig. 4): The activation enthalpy of diopside liquid is much larger than that of forsterite and enstatite liquids. This appears to be odd, inconsistent with FPMD prediction of similar activation enthalpies for all three liquids. Some comments will be helpful.

f) Anomalous viscous behavior (that is, decreasing viscosity with increasing pressure) is clearly observed by this study (Figure 2). The authors need to further discuss this behavior.

g) Correlations between melt viscosity and density, and interpretation in terms of structural changes as discussed here appears to be mostly fine. However, changes in various mean coordination involving cations and oxygen may not be sufficient. For instance, it may be highly abundant five-fold Si-O coordination species/states which causes relatively weak or anomalous pressure variations of viscosity of silicate melts. The authors need to talk about this possibility as well.

h) Implications of the measured viscosity data of three liquids for

chemical stratification following the solidification of magma ocean are interesting. However, further elaborating on viscosity results themselves would strengthen the paper. For instance, either extended Table 3 or extended Fig 4 talking about activation enthalpies should appear in the main body of the paper. Similarly, some parts of extended discussion should go to the main text.

Reviewer #3 (Remarks to the Author):

This manuscript reports new and state-of-the-art measurements on viscosities of a number of end-member silicate liquids. This is a major experimental achievement, which essentially doubled the pressure capability and increased temperature range by ~50%. The approach has the potential for much improved experimental constraints on physical properties of silicate liquids and fractionation models for mantle evolution.

Unfortunately, the manuscript does not give deserved credit to such a technological and experimental feat. It puts main emphasis on implications of the new results, with numerous hidden assumptions which are not properly justified. To me, it reads like two papers put into one. This may be largely due to the current Nature format. For papers in Nature Communications, they can actually expand the main text significantly, with a larger number of display items (up to 10?) and use sub-headers for different topics. I recommend the manuscript, but not in its current format. It must be significantly rewritten, following Nature Communications guidelines, and address the following issues.

Major issues

Experimental:

1. Possible Re in silicate liquids. 3000 K is very high temperature where all elements are highly mobile. Need probe data to check possible Re presence in quenched liquids.
2. Although FEA gave a temperature gradient of ~20 K, a proof of such small gradient should be demonstrated experimentally. An SEM sample section should be shown. Quenched crystal sizes and distribution may have information on actual temperature gradient in the sample. The standard technique of geothermometer (by mapping composition variations of a solid phase assembly) is another possibility.
3. For terminal velocity determination, simply fitting part of the falling distance by a straight line is not accurate enough. You should take local derivatives of falling distance with respect to time and plot such velocity as a function of falling distance. This way, you can determine the distance interval where velocity is at maximum and avoid potential bias by accidentally including lower velocity segments.
4. Viscosity uncertainty of 6% seems extremely small. I don't see any details of the Monte Carlo simulation anywhere in the manuscript. Correction factors in Faxen's calculations are controversial and the wall effect correction factors alone may have errors greater than 10%. End effects may be of similar magnitude. See, for example, Fidleris and Whitmore, British Journal of Applied Physics, 12(9), 490-494, 1961. The correction factors also depend on the Reynolds number (viscosity and density of the liquid, as well as falling speed) and the ratio of diameters of the sphere and the sample chamber.

At very high pressures and temperatures, EoS based estimations on probe sphere diameter and density will also have significant errors. Factors like these must be carefully discussed when presenting uncertainties of viscosities.

Modeling:

1. Liquid densification mechanism and Extended Data Fig. 3. This discussion sounds quite sketchy. The MD simulation for coordination change (ref 40) is for sodium silicate, where Na is a well-known network modifier, at least at low pressures. We also know that coordination change in silicate melts depends on composition. Al-rich melts tend to increase coordination number at lower pressures, as shown in Ref [41]. Effects of FeO/Fe₂O₃ on viscosity are not discussed at all. As fractionation occurs, how is Fe partitioned between liquid and solid, and how would that affect density calculation?
2. Also, the entire calculation assumes completely dry system with no water. This should at least be mentioned.
3. The Adam-Gibbs theory as stated in Equation (10) does not take proper account of elemental partitioning among the four liquid components involved. For example, enstatite liquid may take large amounts of Al₂O₃, competing with anorthite liquid. As the composition of each component changes, viscosity of that component will also change.
4. ED Fig. 4: fitting the enthalpies using linear relations is poorly justified. As shown in EDF 3, all of the liquids undergo transformation with increasing coordination. Liquids in the lower mantle are different in coordination from those near the surface and in the upper mantle for any given composition. Based on ED Fig. 3. it makes more sense to use the data above ~20 GPa to extrapolate to the deep lower mantle. This highlights the uncertain nature of the calculation results.

Minor:

Line 24: "a layer of bridgmanite enriched" needs to be fixed.

Line 33-36: awkward sentence

Line 42: "besides" with an "s"

Line 107: "satisfactorily"

Revision information

First of all, we sincerely appreciate all the three reviewers for their evaluation of our manuscript (NCOMMS-19-14078-T) entitled “Viscosity of the Magma Ocean and the primordial structure of Earth’s silicate mantle”. Comments of the reviewers have been very useful to improve our manuscript significantly.

The following is a list of our revisions:

Reviewer #1’s editorial comments to Author (our responses are in blue font, citation from the main text is in brown)

This study reports novel, highly accurate in situ measurements of the viscosity of silicate melts at extremely high pressures and applies the results to the early molten Earth, concluding that crystal/melt segregation in the molten mantle and the associated chemical differentiation was limited to the uppermost part of the lower mantle and the upper mantle. The type of experiments that the authors report are among the most difficult ones to do. This is clearly the strongest part of the study.

The application part could be somewhat better. For example, during crystallization of the shallow parts of the mantle the crystal size is controlled not by nucleation but by Ostwald ripening - the switch in the controlling mechanism occurs when the temperature drops below liquidus at all depths (ref 5). The crystals can grow an order of magnitude larger. This does not change the main conclusions but it does change the confidence level and the wording of the conclusions - in the upper mantle, fractional crystallization not just “could have also occurred” (last sentence of the abstract), but almost certainly occurs, even regardless of the heat flow uncertainties.

Thanks for the constructive comment. We now consider the effect of Ostwald ripening on grain growth and the change of controlling mechanism from nucleation to grain growth in lines 178-196, 424-472. In particular, we re-calculated R_{cc} values including this effect in Fig. 4 and Supplementary Fig. 11.

For the sake of completeness of error analysis, the authors should use some errors in the formulas describing the critical crystal size, convective velocities, etc, in their theoretical formulas and fig. 4.

Following the suggestion, we added error analysis.

- concerning temperature at the surface of the MO (lines 397-401):

“For a moderately opaque atmosphere with $C_f=0.75$, the heat flux is ~3 times lower than for $C_f=1$. T_{sur} is related to the potential temperature (T_{poten}) of MO through the scaling law. When T_{poten} is 2000 K, T_{sur} is ~ 1800 K and 1400 K for hard and soft turbulent convection, respectively⁵. Here, we assume the T_{sur} equals T_{poten} . Therefore, T_{sur} is overestimated by 1.1 to 1.4 times.”

- Concerning relative velocity between crystal and melt (line 404):

“The overestimation of T_{sur} causes an overestimation of v_s by ~1.4 to 4 times”

- Concerning critical size (lines 422-423):

“Because we overestimate the heat flux by ~1.4 to 4 times, the calculated value of dc is at least overestimated by ~2 or 14 times.”

- Concerning crystal size (line 448-451; 471):

“Therefore, the crystal size is mainly controlled by its nucleation size, when the crystallization zone is covered by a fully-molten layer. In such conditions, we can ignore grain growth and Eq. (25) provides a lower limit for the crystal size in the crystallization zone.

“The lower limit of crystal size can be expressed as:”

- Concerning R_{cc} value (lines 479-481):

“Since our model overestimates critical size (d_c) by ~2 or 14 times and uses a lower limit of crystal size ($d_{crystal}$), it also underestimates the R_{cc} value and favors equilibrium solidification.”

The word “entrainment” on line 151 assumes a mechanism of suspension which is not required by formula 11. “Suspension” is physically a more accurate term.

Following your suggestion, the word “entrainment” was changed to “suspension” in lines 148 and 478.

Reviewer #2 (Remarks to the Author):

The paper by Jie et al. reports new experimental data on pressure variation of viscosity of three key liquids: forsterite, enstatite and diopside. It also presents implications for magma ocean dynamics and cooling.

In overall, the work is interesting, significantly advancing our knowledge of viscous behavior of silicate melts under relevant P-T conditions.

a) It may not be true that the calculated melt viscosities differ by factor of 50 or so between different computational studies. Actually, first-principles molecular dynamics (FPMD) simulations are robust and reasonably accurate. FPMD results are within a factor of 2 or 3 of existing experimental data at zero or low pressures for all three (forsterite, enstatite and diopside) liquids studied in this study (please see Ref. 7 and 15). So, one should not put FPMD study in the same category as empirical model calculations such as Ref 6. The authors should rephrase lines 45 to 48 to avoid this confusion and refer to relevant references. One such reference is: Verma and Karki, First-principles study of self-diffusion and viscous flow in diopside (CaMgSi₂O₆) liquid, *Am. Mineral.* 97, 2049, 2012.

Following the suggestion, we rewrote the description of computational studies in lines 42-51:

“and first-principles and empirical molecular dynamics simulations present a large discrepancy. For example, viscosities differing by a factor of 50 were reported at the lowermost-mantle P-T conditions of 120 GPa and 4000 K^{e.g.6,7}. First-principles molecular dynamics (FPMD) calculations should be more robust than empirical molecular dynamics simulations, because of absence of assumption about the charge density. They provide viscosity values within a factor of 2 or 3 of experimental data obtained at low pressures and may have an advantage for simulations at very high pressures⁷⁻⁹. However, experimental measurements are critically needed to confirm calculations and refine viscosity values, especially at lower mantle P-T conditions.”

The reference (ref. 9) was added in line 49.

“Verma and Karki, First-principles study of self-diffusion and viscous flow in diopside (CaMgSi₂O₆) liquid, *Am. Mineral.* 97, 2049, 2012.”

b) The paper talks about molecular dynamics simulation study in the context of densification mechanism. The reference is missing.

Following the suggestion, the reference (ref. 22) is added in lines 104,106.

“Noritake, F., & Kawamura, K. Structural transformations in sodium silicate liquids under pressure: A molecular dynamics study. *J. Non-Cryst. Solids* **447**, 141-149 (2016).”

c) It would be helpful to show the first-principles results along 3000 K for comparison in Figure 2 (and Extended Fig. 2). These results are available for each of three melts studied in this paper (see Ref. 7 and 15 for enstatite and forsterite, and Verma and Karki, 2012 for diopside). The measured relatively weak pressure dependence of melt viscosity appears to be consistent with computational trends.

Thank you for the constructive suggestion.

The first-principle results were added in Fig. 2 and Supplementary Fig. 2.

Discussion on comparison between FPMD results and our experimental data was also added in lines 97-98:

“Our experimental results are quite consistent (within one order of magnitude) with FPMD predictions, especially for En and Fo composition (Fig. 2).”

d) Viscosity results are available for multi-component melts (of similar or different) compositions over pressure regimes of upper mantle and shallow lower mantle. Their viscosity might also show modest variations with depth like this study. The authors show (or at least, discuss) some relevant results (Figure 4). Please see the following references:

- Wang et al., Atomistic insight into viscosity and density of silicate melts under pressure. *Nat. Commun.* 5, 3241, 2014.

- Karki et al., Simulation of silicate melts under pressure, *Magma Under Pressure*, (Chapter 16): 419, 453, 2018.

Following the constructive suggestion, the discussion of relevant results was added in lines 100-105.

“En melt and, to a lesser extent, Di melt show a negative pressure dependence in some pressure ranges. Such an anomalous behavior was also reported in a basalt and another silicate melt¹⁸, based on both FPMD simulation¹⁹ and experimental measurements²⁰⁻²¹. The negative pressure dependence is due to either the Si-O bond weakening by the pressure-induced bending of the Si-O-Si angle²¹⁻²² or possibly the increasing concentration of five-fold Si-O coordination species²³⁻²⁴.”

Ref. 18-21 were added.

“Suzuki, A. et al. Viscosity of albite melt at high pressure and high temperature. *Phys. Chem. Miner.* 29(3), 159-165 (2002).

Karki, B.B. et al., Simulation of silicate melts under pressure, In *Magma Under Pressure*, 419-453, Elsevier (2018).

Sakamaki, T. et al. Pondered melt at the boundary between the lithosphere and asthenosphere. *Nature Geosci.* 6(12), 1041 (2013).

Wang, Y. et al. Atomistic insight into viscosity and density of silicate melts under pressure. *Nature commun.* 5, 3241(2014).”

e) Activation enthalpy (Extended Table 3 and Fig. 4): The activation enthalpy of diopside liquid is much larger than that of forsterite and enstatite liquids. This appears to be odd, inconsistent with FPMD prediction of similar activation enthalpies for all three liquids. Some comments will be helpful.

Discussion on activation enthalpies was re-written in lines 113-120 as:

“The refined E_a values at room pressure are ~100 and 150 kJ/mol for Fo, and En melts, respectively, in agreement with previous FPMD predictions^{7,8}. Di melt presents a relative large E_a (250 kJ/mol), which is consistent with diffusion experiments (268 kJ/mol)¹⁷ but larger than FPMD prediction (148 kJ/mol)⁹. Further work is needed to solve this discrepancy. However, it will not affect the conclusion of this article significantly because (i) the amount of diopside is less than 2% in relevant mantle compositions (Table 4) and (ii) only extrapolations to the highest mantle pressures could be affected by significant uncertainties, while viscosity at moderate pressures are satisfactorily constrained by the present results.”

f) Anomalous viscous behavior (that is, decreasing viscosity with increasing pressure) is clearly observed by this study (Figure 2). The authors need to further discuss this behavior.

Following the suggestion, discussion was added in lines 100-105.

“En melt and, to a lesser extent, Di melt show a negative pressure dependence in some pressure ranges. Such an anomalous behavior was also reported in a basalt and another silicate melt¹⁸, based on both FPMD simulation¹⁹ and experimental measurements²⁰⁻²¹. The negative pressure dependence is due to either the Si-O bond weakening by the pressure-induced bending of the Si-O-Si angle²¹⁻²² or possibly the increasing concentration of five-fold Si-O coordination species²³⁻²⁴.”

g) Correlations between melt viscosity and density, and interpretation in terms of structural changes as discussed here appears to be mostly fine. However, changes in various mean coordination involving cations and oxygen may not be sufficient. For instance, it may be highly abundant five-fold Si-O coordination species/states which causes relatively weak or anomalous pressure variations of viscosity of silicate melts. The authors need to talk about this possibility as well.

Thank you for the comment. The alternative explanation was added in lines 104-105, 753-756.

“possibly the increasing concentration of five-fold Si-O coordination species²³⁻²⁴.”

“this compression mechanism a negative pressure dependence of the viscosity, due to the bending of Si–O–Si structural units or possibly the increasing concentration of five-fold Si-O coordination species, as suggested by NMR spectroscopy on quenched glasses²⁴ and molecular dynamics calculations²³.”

h) Implications of the measured viscosity data of three liquids for chemical stratification following the solidification of magma ocean are interesting. However, further elaborating on viscosity results themselves would strengthen the paper. For instance, either extended Table 3 or extended Fig 4 talking about activation enthalpies should appear in the main body of the paper. Similarly, some parts of extended discussion should go to the main text.

Thank you for the constructive comment. We moved table 3 to the main text and the related discussions to lines 110-124:

“Extrapolation of melt viscosity to deep lower mantle conditions

The knowledge of the dependence of the melt viscosity along mantle isotherms enables the refinement of the true activation enthalpy and its pressure dependence (E_a in Eq. (1); Supplementary Fig. 5). The refined E_a values at room pressure are ~100 and 150 kJ/mol for Fo, and En melts, respectively, in agreement with previous FPMD predictions^{7,8}. Di melt presents a relative large E_a (250 kJ/mol), which is consistent with diffusion experiments (268 kJ/mol)¹⁷ but larger than FPMD prediction (148 kJ/mol)⁹. Further work is needed to solve this discrepancy. However, it will not affect the conclusion of this article significantly because (i) the amount of diopside is less than 2% in relevant mantle compositions (Table 4) and (ii) only extrapolations to the highest mantle pressures could be affected by significant uncertainties, while viscosity at moderate pressures are satisfactorily constrained by the present results.

Linear fits (Table 1) enable the extrapolation of the melts viscosity towards the very deep mantle using the Arrhenius equation. They average the complex pressure-dependence of viscosity at low pressures. Still, the fitted E_a values remain within 10 % of the experimental E_a (Supplementary Fig. 5). ”

Reviewer #3 (Remarks to the Author):

This manuscript reports new and state-of-the-art measurements on viscosities of a number of end-member silicate liquids. This is a major experimental achievement, which essentially doubled the pressure capability and increased temperature range by ~50%. The approach has the potential for much improved experimental constraints on physical properties of silicate liquids and fractionation models for mantle evolution.

Unfortunately, the manuscript does not give deserved credit to such a technological and experimental feat. It puts main emphasis on implications of the new results, with numerous hidden assumptions which are not properly justified. To me, it reads like two papers put into one. This may be largely due to the current Nature format. For papers in Nature Communications, they can actually expand the main text significantly, with a larger number of display items (up to 10?) and use sub-headers for different topics. I recommend the manuscript, but not in its current format. It must be significantly rewritten, following Nature Communications guidelines, and address the following issues.

1. Possible Re in silicate liquids. 3000 K is very high temperature where all elements are highly mobile. Need probe data to check possible Re presence in quenched liquids.

Following the suggestion, we added EPMA data in supplementary table 2, in which the Re content is found around the detection limit. Following are our arguments on the possible Re contamination issue:

- The Re sphere takes only a fraction of second to fall after the sample is molten, which is extremely short for any Re to diffuse from the sphere to the silicate melt.
- There is no detectable variation of the sphere size during its fall, based on the recorded sample images. Due to the spatial resolution of ~2 μm , the change of sphere diameter in our experience is less than ~4 μm . Considering a sphere of 70 μm diameter, the calculated Re content in melt should be less than 150 ppm, which is under the detection limit of the EPMA.
- We note that potential traces of Re in the recovered samples would be more likely to arise after the sphere falls, when we determine the pressure at high temperature performing X-ray diffraction in the MgO pressure medium. This step can take several minutes, which is 2 or 3 orders of magnitude longer than that of the sphere falling process.

2. Although FEA gave a temperature gradient of ~20 K, a proof of such small gradient should be demonstrated experimentally. An SEM sample section should be shown. Quenched crystal sizes and distribution may have information on actual temperature gradient in the sample. The standard technique of geothermometer (by mapping composition variations of a solid phase assembly) is another possibility.

Following the suggestion, we add a SEM image in Supplementary Fig. 1c. The distribution of crystal size is homogeneous in the whole capsule, and may suggest a very homogeneous temperature. Also, we want to emphasize that it is an indirect method and highly affected by the quench speed and cooling history of the sample.

Common geothermometers are calibrated under low pressure (<16 GPa, such as Westrenen et al. 2003) and maybe unreliable at higher pressure due to phase transitions.

Instead, we developed a diffraction method to measure in situ the actual temperature gradient in this study, see in the method section (lines 281-286) and Supplementary Fig. 1d.

“To estimate temperature gradients, we recorded diffraction patterns at different positions in the MgO pressure marker (positions noted P1, P2, and P3 in Supplementary Fig. 1d). Under the assumption of a negligible pressure gradient inside heater, the difference in MgO volumes can be attributed to a temperature difference. At a thermocouple temperature of 1273 K, the resulted temperature difference is less than 60 K between the thermocouple and center of capsule, and less than 10 K between the center and the top of capsule.”

3. For terminal velocity determination, simply fitting part of the falling distance by a straight line is not accurate enough. You should take local derivatives of falling distance with respect to time and plot such velocity as a function of falling distance. This way, you can determine the distance interval where velocity is at maximum and avoid potential bias by accidentally including lower velocity segments.

In our work, the terminal velocity was determined in the same way as the reviewer proposes. The velocity-time diagram was added in figure 1c together with the following figure caption: “c, Velocity/time plot of the sphere in Run MA24, using a sampling time of 10 ms. The red dashed line is a best fit through the data points located on the "velocity plateau" corresponding to the terminal velocity.”

4. Viscosity uncertainty of 6% seems extremely small. I don't see any details of the Monte Carlo simulation anywhere in the manuscript. Correction factors in Faxen's calculations are controversial and the wall effect correction factors alone may have errors greater than 10%. End effects may be of similar magnitude. See, for example, Fidleris and Whitmore, British Journal of Applied Physics, 12(9), 490-494, 1961. The correction factors also depend on the Reynolds number (viscosity and density of the liquid, as well as falling speed) and the ratio of diameters of the sphere and the sample chamber. At very high pressures and temperatures, EoS based estimations on probe sphere diameter and density will also have significant errors. Factors like these must be carefully discussed when presenting uncertainties of viscosities. This comment contains two parts: 1. Monte Carlo simulation and 2. Faxen's calculations.

1. Following the suggestion, the details of the Monte Carlo simulation was added in lines 323-334 and Supplementary Fig. 3.

“Error analysis and reproducibility

We conducted Monte Carlo simulations to evaluate the propagation of experimental uncertainties on pressure, temperature, terminal velocity and sphere size. Gaussian distribution of experimental uncertainties was assumed. The sampling number was 10000. The results for Run S3219 (En, 24.1 GPa, 2836 K) are presented as an example (Supplementary Fig. 3). Even though the relative uncertainties of pressure and temperature are larger (~3.6%) than those of terminal velocity (0.5%), their contribution to the final viscosity is 1 order of magnitude smaller. This is because the density contrast between sphere and melts is not sensitive to pressure and temperature. The main source of uncertainty for the final determination of the melt viscosity is caused by the uncertainty of 2.9% on the sphere diameter, which is elevated to a quadratic power in the expression of viscosity. The total error on viscosity is within 6%, with an almost Gaussian distribution (Supplementary table 1).”

We also note that we performed repeated experiments to check the reproducibility (lines 335-338).

“Reproducibility of our measurements was checked by performing repeated experiments at similar pressures, temperatures and with different sphere sizes (such as run S3170 and S3171, S3172 and S3175, S3257 and S3260; see Extended table 1). The difference between repeated experiments remains within 6%, which is consistent with the estimated viscosity error.”

2. On the correction factors in Faxen's calculations:

There are discrepancies among different models of correction factors, especially for large ratios of $r_{\text{sphere}}/r_{\text{chamber}}$ and in different flow regimes (Fidleris and Whitmore, 1961). The different models seem to converge when $r_{\text{sphere}}/r_{\text{chamber}}$ is small and in the same flow regime. The Reynolds numbers of all our experiments (0.01-0.1) are far smaller than 1, which points out a laminar flow regime. In our experiments, the $r_{\text{sphere}}/r_{\text{chamber}}$ is ~0.13, which suggests a difference between empirical and predicted model of less than 3% (Fidleris and Whitmore,

1961) and a difference among empirical models (such as Faxen's, 1922 and Francis, 1933) of less than 1%.

Modeling:

1. Liquid densification mechanism and Extended Data Fig. 3. This discussion sounds quite sketchy. The MD simulation for coordination change (ref 40) is for sodium silicate, where Na is a well-known network modifier, at least at low pressures. We also know that coordination change in silicate melts depends on composition. Al-rich melts tend to increase coordination number at lower pressures, as shown in Ref [41]. Effects of FeO/Fe₂O₃ on viscosity are not discussed at all. As fractionation occurs, how is Fe partitioned between liquid and solid, and how would that affect density calculation?

This comment contains three topics: 1. Liquid densification mechanism, 2. Al and Fe effect on viscosity calculation and 3. Effect of Fe partitioning for density calculation.

1. Liquid densification mechanism:

In order to understand the densification mechanism, the ideal way is to obtain the structure of three liquids from MD simulation or experiments (very difficult). However, there is no such results in literature. Because sodium-silicate shows many different densification mechanisms at small pressure range (ref. 22). We used it as an analogue to understand the possible densification mechanisms and their characteristics, summarized in lines 745-761.

“The relatively complex pressure dependence measured in this study (Supplementary Fig. 4) can be related to the densification mechanisms. Based on molecular dynamic simulation²², three distinct densification mechanisms (T1, T2 and T3) were proposed before the coordination number of Si changes upon compression of sodium silicate melts. At the lowest pressures (T1 mechanism), silicate melts behave like ionic liquids consisting of glass-structure modifier cations (Mg or Ca) and SiO₄ groups; The main densification mechanism is the change in coordination number (CN) of modifier cations and this is expected to induce an increase of the melt viscosity with increasing the pressure. Within the T2 region, the main densification mechanism is the collapse of the SiO₄ network; this compression mechanism a negative pressure dependence of the viscosity, due to the bending of Si–O–Si structural units or possibly the increasing concentration of five-fold Si–O coordination species, as suggested by NMR spectroscopy on quenched glasses²⁴ and molecular dynamics calculations²⁵. This compression mechanism is also accompanied by a continuous, progressive, change of the CN of modifier cations. Then, in the T3 region, the silicate melt structure gradually evolves toward a coesite-like network through an increasing number of four-membered tetrahedral rings and a decreasing number of five to seven-membered rings; It yields an increase of the melt viscosity with increasing pressure, while the cation CN remains almost constant.”

Even though the composition will affect the pressure range of a densification mechanism, it should not affect the characteristics of densification mechanism. It should be quite robust to apply the characteristic of densification mechanism to identify the densification mechanism of Fo, En, Di melts, summarized in 762-768.

“Altogether, the pressure dependence of the viscosity of Fo, En and Di melts can be interpreted based on the progressive evolution between these three densifications mechanisms (Supplementary Fig. 4a, b, and c). They are T2 (0 to ~10 GPa) and T3 (10 to 30 GPa) for Fo-melt; T2 (0 to ~10 GPa) and T3 (~10 to ~30 GPa) for En-melt and T1 (0 to ~5 GPa), T2(~5 to ~21 GPa) and T3(~21 to ~30 GPa) for Di-melt. Comparison between En and Di melts suggests that higher pressures are required to induce the change of densification mechanisms from T1 to T2, and T2 to T3, when the modifier-cation is larger (Supplementary Fig. 4d).”

2. Al and Fe effect on viscosity calculation

We used Adam-Gibbs theory for compositional extrapolation of viscosity. Al element is included in the endmember of anorthite. The Al effect on the melt structure is considered during the calculation in the anorthite term of Eq. 10.

The effect of Fe is negligible because of the low concentration of fayalite endmember and small difference between liquid viscosity of fayalite endmember and fosterite endmember. We add discussion on Fe effect in lines 357-362 and add Supplementary fig. 6.

“We chose viscosity function of Fo for both Fo and Fa components. Because Fa component represents less than 8% of the KLB-1 and Chondritic composition and, in addition, viscosities of Fa and Fo converge to the same value (the difference is within ~10%) with increasing pressure (Supplementary Fig. 6). The total error caused by ignoring the Fa component is less than 0.8%.”

We note that our predicted viscosity of peridotite melt is quite consistent with the existing experimental data as shown in Fig. 3a and description in lines 137-139.

“Our calculations for KLB-1 composition (Fig.3a) is roughly compatible with available measurements³⁰.”

3. Effect of Fe partitioning for density calculation

Fe solid-liquid partition coefficient (K_{Fe}) from 0.2 to 0.6 (see ref. 35 and references therein). Following the suggestion, we recalculate R_{cc} value using this range. This does not affect our conclusion significantly. We added description in lines 168-171, 482-497:

“To model the solid-melt density contrast, we consider a range of Fe solid-liquid partition coefficient (K_{Fe}) from 0.2 to 0.6 (see ref. 35 and references therein) and first liquidus phases that change with MO depth (see Methods). Higher K_{Fe} favors higher density contrast of bridgmanite over liquid.”

“Density of melt and crystal in MO

On the course of MO cooling, we considered the possible crystallization of bridgmanite ((Mg_{1-x}Fe_x)SiO₃) or ferropericlae (Mg_{1-x}Fe_x)O at pressures above 23 GPa, garnet ((En₈₀Py₂₀)_{1-x}Al_x) between 15 and 23 GPa, and olivine ((Mg_{1-x}Fe_x)₂SiO₄) below 15 GPa. As our results show that crystal fractionation remains limited to some mantle regions, we assume a constant melt composition on the course of the MO solidification. Density of the MO melt was calculated from endmember melt compositions using the ideal mixing model³¹. Iron content in crystals were calculated based on the melt composition and crystal-melt partition coefficients. We considered partition coefficients varying from 0.2 to 0.6, due to remaining experimental uncertainties (see ref. 3 and references therein). To calculate density of bridgmanite, ferropericlae, majoritic garnet and olivine, we used an ideal lattice mixing model⁶⁴ between end-member compositions with the following EoS: bridgmanite ((Mg_{1-x}Fe_x)SiO₃)⁶⁴; (Mg_{1-x}Fe_x)O as a solid solution of MgO⁴⁰ and FeO⁶⁵; En₈₀Py₂₀)_{1-x}Al_x as a solid solution of En₈₀Py₂₀^{ref.66} and Almandine⁶⁷; Ol (Mg_{1-x}Fe_x)₂SiO₄ as a solid solution of (Mg_{0.9}Fe_{0.1})₂SiO₄^{ref.68} and Mg₂SiO₄^{ref.69}.”

2. Also, the entire calculation assumes completely dry system with no water. This should at least be mentioned.

Following the suggestion, we now added the information in lines 133-134:

“Because water has little effect on a completely depolymerized, high temperature magma viscosity²⁸⁻²⁹, we only consider the dry MO.”

3. The Adam-Gibbs theory as stated in Equation (10) does not take proper account of elemental partitioning among the four liquid components involved. For example, enstatite

liquid may take large amounts of Al₂O₃, competing with anorthite liquid. As the composition of each component changes, viscosity of that component will also change.

Our argument is as follows:

The elemental partitioning is an important concept for estimating the bulk properties of solid phases because we can distinguish different phases. However, it does not apply to a mixture between end-members of a melt, because they are intimately mixed with each other (there is no "partitioning", because there is only a single phase: the melt).

4. ED Fig. 4: fitting the enthalpies using linear relations is poorly justified. As shown in EDF 3, all of the liquids undergo transformation with increasing coordination. Liquids in the lower mantle are different in coordination from those near the surface and in the upper mantle for any given composition. Based on ED Fig. 3, it makes more sense to use the data above ~20 GPa to extrapolate to the deep lower mantle. This highlights the uncertain nature of the calculation results.

Interestingly, this Rev. 3 comment #4 is contradictory with Rev. 2 comment #h, who finds our procedure very interesting and requests that either ED Fig. 3 or ED table 3 should be moved to the main text.

We agree that there is more uncertainty when the viscosity is extrapolated at depth >820 km. However, the main implications of our article concern the top lower mantle (from 660 to ~1000 km), which is a range of depths partially covered by our experimental measurements. As an illustration of this, our conclusions are unchanged whether we adopt an extrapolation with or without data below ~20 GPa.

Since we cannot be sure that the compaction of the melt structure is finalized at ~20 GPa, we prefer to refine our Arrhenius equation on the entire range of available experimental pressures.

Minor

Line 24: "a layer of bridgmanite enriched" needs to be fixed.

Following the suggestion, "a layer of bridgmanite enriched" was changed to "a bridgmanite-enriched layer" in line 26.

Line 33-36: awkward sentence

Following the suggestion, we changed lines 32-35 to

"The possibility that the MO induced a primordial chemical stratification has major implications for the mantle state and its dynamics over the Earth's history. For example, it could have induced large-scale provinces atop the core-mantle boundary¹ or a basal MO that would have taken several billion years (Ga) to crystallize²."

Line 42: "besides" with an "s"

Following the suggestion, "beside" was changed to "besides" in line 41.

Line 107: "satisfactorily"

Following the suggestion, "satisfactory" was changed to "satisfactorily" in line 120.

REVIEWERS' COMMENTS:

Reviewer #1 (Remarks to the Author):

I find the changes satisfactory.

Reviewer #2 (Remarks to the Author):

The revision of the paper by Xie et al. has addressed the points/concerns raised by reviewer #2 as well as others. Their measured viscosity data do compare well with previous experimental data and FPMD results. The presentation and analysis of their data and detailed exploration of implications for magma ocean dynamics look impressive. I believe that this revised version should be acceptable for publication.

Reviewer #3 (Remarks to the Author):

This revision is much improved over the previous one. Most of my review comments have been properly addressed and I recommend acceptance. I have only some minor comments, which are listed below.

Line 64: Fig. 1c seems to refer to the main figure, not Supplementary Fig. 1c.

Lines 98-99: The first statement is basically a repeat of Lines 78-79.

Lines 115-118: can you state errors in the E_a values thus determined?

There is an issue with the choice of indefinite articles (e.g., a or an) and definite articles (the) in the manuscript, especially when referring to magma oceans. For example, "the early magma-ocean (MO) solidification" in Line 22 is better to be replaced with "an early magma-ocean (MO) solidification". This applies also to Line 33. Similarly, "the MO" in Lines 40 and 42 should be simply "MO". "the dry MO" in Line 136 should be "dry MOs". In Lines 147, "the MO" should be "an MO"; in Line 150 "the convective MO" should be "a convective MO". In Line 180, "the MO" should be "an MO", and then in the rest of the paragraph, "the MO" will be appropriate. Finally, in Line 225, "the shallow MO" should be "a shallow MO" or "shallow MOs".

Lines 59-62: How about "We used a relatively large beam (...) to record the falling of a small rhenium sphere in the liquid silicate with an ultra-fast camera (Fig. 1) and a collimated beam (...) to characterize sample mineralogy and determine the pressure by X-ray diffraction."

Revision information

First of all, we sincerely appreciate all the three reviewers for their second evaluation and positive response of our manuscript (NCOMMS-19-14078-T) entitled “Viscosity of the Magma Ocean and the primordial structure of Earth’s silicate mantle”.

The following is a list of our revisions:

Reviewer #1 ’s editorial comments to Author (our responses are in blue font, citation from the main text is in brown)

I find the changes satisfactory.

Reviewer #2 (Remarks to the Author):

The revision of the paper by Xie et al. has addressed the points/concerns raised by reviewer #2 as well as others. Their measured viscosity data do compare well with previous experimental data and FPMD results. The presentation and analysis of their data and detailed exploration of implications for magma ocean dynamics look impressive. I believe that this revised version should be acceptable for publication.

Reviewer #3 (Remarks to the Author):

This revision is much improved over the previous one. Most of my review comments have been properly addressed and I recommend acceptance. I have only some minor comments, which are listed below.

Line 64: Fig. 1c seems to refer to the main figure, not Supplementary Fig. 1c.

Following the suggestion, we changed the “Supplementary Fig. 1c” to “Fig. 1c” in Line 72.

Lines 98-99: The first statement is basically a repeat of Lines 78-79.

Lines 78-79 describes the viscosity behaviors along liquidus, the temperature of which varies with pressure.

Lines 98-99 describes the viscosity behaviors along isotherms, which is the pure pressure effect on viscosity.

Therefore, we think it is necessary to keep both statements.

Lines 115-118: can you state errors in the E_a values thus determined?

Following the suggestion, we stated errors in Lines 123-125:

“The refined E_a values at room pressure are 100 ± 20 and 159 ± 10 kJ mol^{-1} for Fo, and En melts, respectively, in agreement with previous FPMD predictions^{7,8}. Di melt presents a relative large E_a (230 ± 30 kJ mol^{-1}), which is consistent with diffusion experiments (268 kJ mol^{-1})¹⁷ but larger than FPMD prediction (148 ± 5 kJ mol^{-1})⁹.”

There is an issue with the choice of indefinite articles (e.g., a or an) and definite articles (the) in the manuscript, especially when referring to magma oceans. For example, “the early magma-ocean (MO) solidification” in Line 22 is better to be replaced with “an early magma-ocean (MO) solidification”. This applies also to Line 33. Similarly, “the MO” in Lines 40 and 42 should be simply “MO”. “the dry MO” in Line 136 should be “dry MOs”. In Lines 147, “the MO” should be “an MO”; in Line 150 “the convective MO” should be “a convective MO”. In Line 180, “the MO” should be “an MO”, and then in the rest of the paragraph, “the MO” will be appropriate. Finally, in Line 225, “the shallow MO” should be “a shallow MO” or “shallow MOs”.

Following the suggestion, “the early magma-ocean (MO) solidification” in Line 25 was replaced with “an early magma-ocean (MO) solidification”.
“the MO” in Lines 36, 156 and 189, were replaced with “a MO”.
“the MO” in Lines 43 and 44 were replaced with “MO”.
“the dry MO” in Line 145 was replaced with “dry MOs”.
“the turbulently convective MO” in Line 159 was replaced with “a turbulently convective MO”.
“the shallow MO” in Line 234 was replaced with “a shallow MO”

In addition, we changed

“the rare gases” in Line 40 to “rare gases”
“the melt” in Line 45 to “melt”

Lines 59-62: How about “We used a relatively large beam (...) to record the falling of a small rhenium sphere in the liquid silicate with an ultra-fast camera (Fig. 1) and a collimated beam (...) to characterize sample mineralogy and determine the pressure by X-ray diffraction.”

Following the suggestion, we changed “We used either a relatively large beam (about 2x2 millimeters) to record the falling path of a small rhenium sphere in the liquid silicate using an ultra-fast camera (Fig.1), or a collimated beam (50x200 μm) to characterized the sample mineralogy and determine the pressure by X-ray diffraction.”

to

“We used a relatively large beam (about 2x2 millimeters) to record the falling path of a small rhenium sphere in the liquid silicate using an ultra-fast camera (Fig.1), and a collimated beam (50x200 μm) to characterized the sample mineralogy and determine the pressure by X-ray diffraction.” in lines 66-70.